# Powder Spreading Mechanism in Laser Powder Bed Fusion Additive Manufacturing: Experiments and Computational Approach Using Discrete Element Method

**DOI:** 10.3390/ma16072824

**Published:** 2023-04-01

**Authors:** Ummay Habiba, Rainer J. Hebert

**Affiliations:** Department of Materials Science and Engineering, Institute of Materials Science, University of Connecticut, Storrs, CT 06269-3136, USA

**Keywords:** additive manufacturing (AM), laser powder bed fusion (LPBF), powder spreading, discrete element method (DEM), ESI software, metal powder, powder bed packing density, particle size distribution, porosity, spreader velocity, particle trajectory

## Abstract

Laser powder bed fusion (LPBF) additive manufacturing (AM) has been adopted by various industries as a novel manufacturing technology. Powder spreading is a crucial part of the LPBF AM process that defines the quality of the fabricated objects. In this study, the impacts of various input parameters on the spread of powder density and particle distribution during the powder spreading process are investigated using the DEM (discrete element method) simulation tool. The DEM simulations extend over several powder layers and are used to analyze the powder particle packing density variation in different layers and at different points along the longitudinal spreading direction. Additionally, this research covers experimental measurements of the density of the powder packing and the powder particle size distribution on the construction plate.

## 1. Introduction

Additive manufacturing (AM) is a cutting-edge manufacturing technique that has lately been adopted by different industries, for example, automotive, aerospace, and biotechnology, to build parts with complex geometries [1,2,3,4], and oftentimes also for prototyping. Different metal additive manufacturing processes have been developed, for example, directed energy deposition (DED), wire deposition using laser or electron beam sources, binder jet printing, electron beam melting (EBM), laser powder bed fusion (LPBF), friction stir additive manufacturing, and new technologies combining some of these techniques. Laser powder bed fusion is used when parts with high geometrical complexity and density near 100% are required [5]. Parts are made in LPBF by creating solid material in patterned layers [6]. The powder is used as a raw material in LPBF, with typical particle sizes ranging typically from about 10 to 60 μm [7]. The deposition of a thin layer of powder on a build plate by a spreader is one of the defining aspects of LPBF [8]. A laser beam moves over sections of the powder bed in each layer, melting the powder in those areas to produce solid layers of 3D components. The build plate that contains the additively manufactured parts then lowers, and a new layer of powder is deposited to be melted onto the previously fused layer [9]. The process of lasering, build plate lowering, and powder spreading is repeated and hundreds to thousands of layers typically make up the final product.

The particle size distribution of the powder has been shown to affect the powder flow behavior and packing density of the powder bed [7,10,11,12]. Variations in the powder bed characteristics such as local areas with above or below-average packing densities can cause defects in the additively manufactured parts. Porosity in as-built additively manufactured parts or even unmelts reduces mechanical strength and fatigue performance [13]. Since the porosity of a sample usually affects its fatigue behavior negatively [14], the powder bed characteristics should affect the overall mechanical performance of additively manufactured parts [10]. However, many other factors influence the mechanical properties of additively manufactured samples and continued efforts are therefore needed to isolate the effects of the powder size distribution and spreading behavior on tensile and fatigue properties. While the powder bed variations have been shown to affect mechanical properties, they could also affect the anisotropy of microstructures and properties of additively manufactured samples. Microstructural anisotropy is a factor that often differentiates additively manufactured from cast samples. Differences in additive manufacturing conditions have recently been demonstrated to even affect the milling behavior of additively manufactured IN718 samples [15] and those differences were linked to anisotropic microstructural features. Variations in powder bed packing density, for example, could imply differences in heat flow conditions that could affect the growth conditions in the melt pool.

Several constraints limit experimental studies of powder flow. Powder size distributions, for example, are difficult and costly to adjust; particle morphologies vary and cannot be controlled; performing experiments under controlled environmental conditions has its own challenges. Numerical simulation modeling has been employed to support and complement experimental tests required to grasp the fundamentals of the AM process. The Discrete Element Method (DEM) has recently gained popularity as a technique for simulating powder flow [16]. The effects of spreader spacing and spreader velocity on powder shear band and flow rate were investigated using DEM by Nan et al. [17]. They discovered that the mass flow rate rose linearly with the spreader gap was linearly dependent on spreader velocity for a period of time, and then did not appreciably increase. DEM was employed by Haeri et al. to analyze the distribution of rod-shaped particles in AM [18]. They demonstrated that higher spreader translational velocities or larger particle aspect ratios increase the powder’s surface roughness and packing density. Using the DEM tool, Mussatto et al. [19] investigated the effects of spreader velocity and layer thickness on powder particle size distribution in AM. They found that lowering the spreader velocity improves the homogeneity of the powder bed. Haeri used the DEM tool to evaluate the influence of spreader type on powder bed quality [20]. The super elliptical spreader blade form was discovered to produce a better powder bed quality than other spreader blade shapes.

In this work, the DEM simulation tool is used to investigate the effects of different powder spreading parameters on the powder density and particle distribution within the powder bed. The DEM simulations extend across multiple layers in order to determine changes in powder bed characteristics with the number of layers. Furthermore, the powder particle packing density variation is assessed at different points along the longitudinal spreading direction. This study also includes experimental measurements of the powder packing density and the powder particle size distribution on the build plate. Powder studies help to improve the homogeneity of powder beds and their packing density, which is expected to improve the quality of additively manufactured parts.

## 2. Methodology

### 2.1. DEM Model

The DEM is an approach that considers each particle in a granular system and that determines force interactions of each particle with surrounding particles or walls to arrive at the kinematics of each particle. The basis of the DEM are Newton’s laws of motion [21]:(1)$$m_{p}\frac{d\overset{⇀}{v_{p}}}{dt}=\sum_{q}\overset{⇀}{F_{pq}}+m_{p}\overset{⇀}{g}$$
(2)$$I_{p}\frac{d\overset{⇀}{\omega_{p}}}{dt}=\sum_{q}\overset{⇀}{T_{pq}}$$

In Equation (1) $m_{p}$ is the particle mass, ${\overset{⇀}{F}}_{pq}$ is the force vector due to the interaction between particle *p*, *q*, and particle-wall interactions, and with $\overset{⇀}{g}$ as the gravity of Earth. In Equation (2) I_p_ is the moment of inertia of spherical particle ‘*p*’, $\overset{⇀}{\omega_{p}}$ is its angular velocity, and $\overset{⇀}{T_{pq}}$ is the total torque on particle ‘*p*’ due to tangential contacts with other particles, rolling, and torsion [22]. The literature distinguishes between hard-sphere and soft-sphere interactions. The simulations in this work were run on commercial DEM software developed by the ESI Group (location: Paris, France; software version: 16.6). Details of the DEM model are summarized in [21]. Table 1 lists the input parameters for the ESI DEM calculations.

In order to run a DEM powder spreading simulation, the geometry must be selected first of the area and layer height that the particles fill. The powder spreading geometry is shown in Figure 1 and the values of the table length, width, displacement, source length, and spreader thickness are given in Table 1. The powder spreading sequence starts with the software generating a powder bed on a source table. The rake then moves the powder from the source table over the table, which in powder bed equipment is typically referred to as the build plate. The term ‘table displacement’ is used to describe the drop in height of the build plate that the additive manufacturing community often regards as the layer height. The software divides up the volume taken up by the build plate area and the table displacement into cells. Figure 2 shows the cell breakdown of the powder bed volume and the coordinate system. For the DEM simulations in this work, the y-direction or spreading direction is divided into 10 units, each with a length of 400 μm; the z-direction is also divided into 10 units, each with a height of 6 μm; the x-direction consists of only one cell with a length of 500 μm. Each cell can be labeled with the coordinates according to the coordinate system shown in Figure 2. The cell highlighted in grey in Figure 2, for example, is the (1,3,2) cell because it is the cell located at one unit in the x-direction, three units in the y-direction, and two units in the z-direction.

The powder spreading direction is along the Y-axis, the additive manufacturing build direction is along the Z-axis, and the direction perpendicular to the spreading direction is along the X-axis. Before the spreading simulation can start, the amount of powder must be determined that is used for the simulation. This amount was calculated using the dosing factor. The dosing factor relates the powder mass used for the simulation to the mass that the volume comprised by the table area times the table height would have if it was filled completely with solid material, i.e., with solid IN718 in this study. For example, a dosing factor of 0.6 indicates that the powder mass used for the simulation is 60% of the mass of IN718 which completely fills the simulation volume of table area times table height. Once the powder mass is known, the total number of powder particles is determined from the known size distribution of the powder particles, and the size distribution is divided into size bins. The particles are distributed in each bin according to the relative distribution of particles according to the size distribution and the total number of particles. The DEM process starts with a heap of powder on the source table. This heap is generated with a gravity-driven drop of the particles from a virtual box that is located above the source table. This box is filled with particles such that the largest particles in the distribution are placed first randomly within the box volume, followed by the second largest particles, etc. After a brief time delay, the spreader starts time, which aims to settle all particles on the source table, the spreader then pushes the powder heap from the source table onto the table. The term “spreader” is used in this work, but sometimes “recoater” is used in the literature. For this study, the spreader was a rectangular box and elastic deformation of the spreader was not considered.

### 2.2. Experimental Methods

Experiments supporting the DEM simulations were performed with IN718 powder with a nominal powder size distribution between 15 µm and 53 µm. The powder size distribution is shown in Figure 3. This size distribution was measured using a Retsch Camsizer XT system. Particle sizes are deduced from projections of powder particles and analyses with high-speed CCD cameras. The powders used for the measurement were accelerated with compressed air through a nozzle in order to break up any agglomerates that might exist in the starting powder. Roughly five million particles were measured for the graph shown in Figure 3. The figure shows the derivative of the cumulative volume-based size distribution as a function of the area-equivalent powder sphere diameter. IN718 is a superalloy with a density of about 8.19 g/cm^3^ and a Young’s modulus value at an ambient temperature of about 200 GPa. Scanning electron microscopy images of individual powder particles are shown in Figure 4a,b. The larger particles appear to be spherical with relatively few satellites. The particle surfaces reveal solidification signatures of dendrites that reached the particle surface during solidification. Satellite particles are also relatively smooth with a spherical to slightly faceted surface morphology. Raking experiments were performed with a manual spreader shown in Figure 5. An exchangeable blade is attached to a rake that can be adjusted at both ends with micrometer dials to a specific height above the aluminum build platform. The rake is mounted on a board that has a rod bearing on both ends so that the board and rake can roll along two parallel rods that are mounted onto the two sides of the build platform. Within the build platform is a circular build plate that can be lowered with another micrometer dial. The rake and blade are moved over the build plate manually. Different blade geometries and types can be used, including commercial blades used for EOS M270 and Arcam A2X additive manufacturing machines.

The powder bed density can be measured following the approach described in [23]. Cylinders were additively manufactured with the design from [23] near the edges and at the center of the build plate of an EOS M270 machine as shown in Figure 6. After removal from the build plate, a small hole is punched into the surface of a cylinder to remove the unmelted powder from the cylinder interior. After recording the powder mass, the empty cylinder is backfilled with water and the amount of water is measured to reflect the volume of empty space inside the cylinder that was taken up by the powder. The ratio of the powder mass and the volume then yields a density of the powder bed. If that density is divided by the solid density of the material—IN718 for this work—a relative packing density is obtained. This relative packing density can then be used further to determine porosity, which is obtained as one minus the relative packing density.

## 3. Results and Discussion

This section is organized as follows. The powder pile is considered the first that forms in additive manufacturing machines when powder drops from hoppers or bins onto the build platform. The initial recoater motion moves this powder pile a short distance before the powder starts to move over the build plate and the powder motion up to the point where powder starts to fall onto the build plate is covered in Section 3.1. Details of powder spreading over the build plate are covered in Section 3.2. In Section 3.2.1 the average powder particle size and size distribution are analyzed along the spreading direction over the build plate. The packing density is then covered in Section 3.2.2 while the final Section 3.2.3 examines the effect of increasing recoater velocity on the packing density.

### 3.1. Initial Powder Configuration and Pileup at Spreader

In any powder bed fusion process, the additive manufacturing machines prepare a small amount of powder for each layer that a raking or roller system then spreads out over the build plate. Two main approaches serve to prepare powder for spreading; either powder is stored in a container and is lifted with the help of a feed piston to be scraped off the top, or powder flows through hoppers and bins and is mostly gravity-driven into the build chamber. The latter powder delivery matches the DEM simulation approach used for this study since both powder deliveries are based on a gravity-driven, “raindrop” seeding of the initial powder heap. The powder heap is then spread out over the build plate and potential irregularities in the arrangement of powder particles in the starting heap before spreading should cause at least some irregularity in the powder arrangement of the powder layer on the build plate. It is therefore important to identify the level of uniformity or variations thereof in the starting powder heap before it is spread over the build plate.

The simulation results shown in Figure 7a depict the surface of the powder after the software dispensed powder onto the source table while Figure 7b depicts the underside of the power. Several particles are visible in Figure 7b that are blue in color and therefore about 10–15 µm in diameter. On the upper side of the powder, only a few blue particles are visible while several red particles are visible. The red particles are among the largest particles of the powder size distribution with a diameter of about 50 µm. Large particles at the top of the powder and small particles at the bottom suggest a size segregation in the starting powder bed before spreading. The vertical size segregation observed in the current DEM simulation for the spreader table has been attributed in the literature to the ability of small particles to move into inter-particle spaces of larger particles when the particle aggregate dilates, for example when poured or vibrated [24].

Once the spreader starts to push the powder heap, the particles pile up in front of the spreader as shown in Figure 8. At the beginning of the spreader motion, the powder heap in front of the spreader does not yet show a clear angle of repose as seen in Figure 8a. With further movement of the rake, the powder pile increases in height and develops a round surface profile as shown in Figure 8b. while a layer of powder on the source table lies still ahead of the rake and the pile. When the rake approaches the build plate, i.e., at the end of the source table, the powder pile starts to pour onto the build plate as shown in Figure 8c. The sequence of powder pile profiles during spreading along the source table is summarized in Figure 9. The powder pile does not develop a single value of the angle of repose, but instead has a concave shape. When the spreader sweeps the particles along the source table, small particles accumulate at the foot of the pile in front of the spreader; Figure 8c shows an accumulation of particles with blue color and therefore diameters of 10–15 µm. The region in the powder pile adjacent to the bottom of the spreader, near the build plate is of particular relevance as shown in [25]. However, the motion of the small particles is only part of the larger question of how the particles form a pile during the raking process. Some of the small (blue) particles are shown in Figure 8c.

### 3.2. Powder Bed

#### 3.2.1. Mean Particle Size and Powder Size Distribution along Spreading Direction

In powder bed additive manufacturing practice, homogeneity of the powder bed across the build plate is thought to be important. Two key characteristics of powder beds are their packing density and their powder size distribution. Both characteristics should be constant across the build plate when the powder is spread over the build plate. The powder bed is homogeneous with respect to the powder size distribution if samples of the powder bed taken at different locations on the build plate yield the same distribution. DEM simulations and experiments were performed to analyze the uniformity of the powder size distribution, the mean particle size at different locations, and the packing density across powder beds. The simulations first only considered a single layer spread over a perfectly smooth build plate and then additional layers spread on top of each other. Qualitative results of the DEM simulation are shown in Figure 10 for the spreading of one layer. The spreader pushes the powder particles first over the source table and then over the build plate, which is lowered relative to the source table by the table displacement of 60 µm. Figure 10 shows that unlike the spreading over the source table, a defined angle of repose develops for the spreading process over the build plate in front of the spreader. During the initial raking stage, from the beginning of the build plate to the center, a 2 mm distance, the dynamic angle of repose is 35–36°. Toward the end of the build plate, at a 4 mm distance, this angle decreases to about 30°. In the literature dynamic angles of repose are measured with the rotating drum technique or from images of powder piles in front of recoaters. Reported values from DEM simulations in the literature are 43° [26], 20.3° for PREP Ti-48Al-2Cr-2Nb powder and 27.6° for gas-atomized powder of the same composition [27], 28.6° in [28]. Mandloi et. al. studied the effects of powder size distributions, cohesive forces between particles, and friction coefficients between particles in DEM simulations of the powder pile in front of a rake [29]. The dynamic angle of repose obtained from the DEM simulations varied approximately between 24°and about 32°. The rolling friction coefficient was found to raise the dynamic angle of repose as did the addition of Van der Waals forces between particles. The effect of cohesiveness on the dynamic angle of repose was significant, but a clear trend with an increase in cohesiveness could not be identified. However, with added cohesiveness between the particles, the dynamic angle of repose could amount to over 40°.

The result of the powder spreading of one layer over the starting plate is shown in Figure 11. The DEM simulation result shows a qualitative size segregation in the spreading direction with the larger particles located toward the end of the 4 mm raking distance and the smaller particles toward the beginning. The DEM simulation was then used to highlight the trajectories of individual particles in the powder pile during the spreading process.

The DEM simulations can be used to identify the trajectories of individual particles or groups of a few particles. Figure 12a shows a sequence of the rake pushing a powder pile over the source table. Forty-three particles are shown in Figure 12a. A close inspection of the particle pile up in front of the rake as shown in Figure 12b shows a tendency of larger particles to move in the powder heap upwards while smaller particles stay approximately at the same height in the heap or trickle slightly down.

The upward motion of larger particles in the powder heap in front of the rake is also shown in Figure 13. The three individual images in Figure 13 show the motion of a 49 µm particle at three different initial locations over the source table and build plate. The particle located initially toward the left end of the source table moves upward during raking and is then moved off the build plate. If the initial location is at the center of the source table or towards the border with the build plate, the particle does gain some height before being deposited on the build plate.

The mean powder particle size on the build plate was analyzed quantitatively using data on the mean particle size in each cell of the powder bed volume. The number of particles analyzed per cell is approximately 300–500. An average was taken of the mean particle sizes in each of the 10 cells in the z-direction for each cell step in the y-direction. The result is shown in Figure 14 and confirms the qualitative observation from Figure 11 that the mean particle size increases along the spreader direction. The increase is highly irregular, which could be caused by the relatively small number of particles in the analysis. The increase in the mean particle size along the spreading direction must be seen in the context of the larger particles moving upward during the spreading process. Figure 13a suggests that this upward motion affects the trajectory and final location of the larger particles, which appears to be shifted to the right at the build plate.

The single powder layer can also be analyzed with respect to the mean particle size in the z-direction. At any cell height along the z-direction, the mean particle sizes were averaged over all x-y cells. The result is shown in Figure 15.

The analysis of the mean particle size as a function of the height in the powder bed, averaged over the build plate area, shows a nearly linear increase in the mean particle size from the bottom of the powder bed to a height of about 30 µm above the bottom, i.e., at the half-height of the powder bed thickness. In the upper half of the powder bed, the average particle size drops with increasing elevation from the bottom of the table. Only the smallest particles are observed near the top of the powder bed. The analysis considers those particles in any cell whose center is located within the cell. The different mean particle diameters are plotted in Figure 16 to scale as a function of their location along the z-axis, i.e., the table displacement.

For the five cells at the bottom of the powder bed at −57 µm, −51 µm, −45 µm, and −39 µm, the mean particle diameters approximately touch the build plate. The largest particle in Figure 16 is roughly the same size as the largest particle in the starting powder size distribution. Since the particles cannot extend into the build plate, the maximum in Figure 15 cannot occur at depths less than the observed cell at −33 µm. The largest particles would be likely raked off the build plate if they were located higher in the powder bed than at the current location of −33 µm in Figure 15. The maximum in Figure 15 is therefore a “geometrically necessary” maximum while the nearly linear drop in mean size toward the bottom of the late is equally geometrically dictated if the particles with the mean sizes indicated in the plot in Figure 15 rest on the build plate. Toward the top of the powder bed, none of the particles in the plot in Figure 16 extend beyond the zero level which represents the motion of the bottom of the rake blade over the build plate. If particles would extend beyond the zero level it can be postulated that these particles would be pushed with the spreader and its blade off the powder bed. The distribution of particles with mean diameters for z-levels from −3 μm to −27 μm must represent a topological way for these particles to be distributed on the particles that rest on the build plate.

The DEM simulations were then extended to successively more layers, resulting in a total of nine layers of powder (Figure 17). Each layer was spread at the same velocity of 0.1 m/s and the same table displacement of 60 µm for each layer onto the previously spread layers. For each layer spreading, all previously spread layers were taken into account in the DEM simulations. The surface profile of the nine-layer powder bed is rather unique: at the beginning of the build plate, the height of the nine layers is greatly reduced. This low powder bed height must be seen in the context of the significant porosity that is observed at the start of the build plate. Fewer particles accumulate in this section than in the remainder of the powder bed. The powder bed height then rises to a maximum before gradually decreasing in height toward the end of the build plate. This gradual decrease in height of the nine-layer powder bed is at least partially due to an increase in the packing density as the spreading distance increases.

Unlike in an experiment, the DEM simulation can analyze the powder size distribution of any individual layer. The powder particle size distribution was nearly identical for layers three to nine as shown in Figure 18. The size distribution in each layer captures all particles on the build plate that were spread over previous layers. Only layers one and two deviated slightly from the distribution of the other layers. The size distribution of the starting powder was modeled after the experimental size distribution shown in Figure 3 for IN718. The size distributions beyond the first two layers, therefore, agree with the starting size distribution. However, within a layer and as a function of the spreading distance, the particle sizes de-mix, as shown in Figure 14.

A manual powder spreading system was employed, and powder was manually distributed over a spread table by moving the straight stainless-steel spreader by hand (Figure 5). The powder was collected and evaluated in the Camsizer analyzer from three different locations on the spread table. The three locations were approximately 150 mm apart, each. In the experimental spreading, the powder layer on the spread table was 200 µm thick and the spreading velocity was approximately 0.1 m/s. Powder samples taken from the three locations were analyzed as shown in Figure 5 and the size distribution at the three locations was identical as shown in Figure 19.

The size demixing that is observed in the DEM simulation along the spreading direction does not contradict the experimental finding. The DEM simulation only covers a distance of 4 mm and it can therefore not be proven or disproven that over a distance of 300 mm, the powder size would remain constant or not. The discrepancy between the experimental analysis capabilities and the DEM simulations cannot be resolved easily. In order to match simulations with experiments, the simulations would have to cover a distance of about 100 mm or more. For shorter distances, the amount of powder that could be extracted from the powder bed at two or three locations along the spreading direction, for example, with tweezers, would not be enough for a statistically sound analysis. However, simulating multiple layers over a distance of 100 mm or more for each layer requires computational capabilities that were not available for this study.

#### 3.2.2. Powder Particle Packing Density

Following the procedure outlined in [23] and described in Section 2, the powder particle packing density varies between 5.0 and 5.1 g/cm^3^ as shown in Figure 20. Three sets of samples were built to obtain an estimate of the standard deviation, which varied between 0.6 g/cm^3^ and 0.01 g/cm^3^. Since the error bars for the locations in the back of the plate, the middle, and the right front overlap, it is not possible to conclude that the packing density varies between those locations. Only the left front location on the build plate has a packing density and error bar that does not overlap with the others and is therefore lower than the density at the other locations. The three builds used different build plates and the same rake blade. The build plate itself should therefore be ruled out as a cause for the lower packing density. However, the build plates rest on a table inside the additive manufacturing machine and it is possible that the lower front table is slightly lowered relative to the other table corners, which would lower the left front plate and therefore increase the volume per powder spread. With a bulk density of IN718 of 8.22 g/cm^3^ [30], the powder bed packing fraction is therefore between 61.2% and 62.4%; the porosity, defined as a 100% dense material minus the observed packing density fraction is therefore between 38.8% and 37.6%. The approach taken in this work to determine the powder bed density is based on additive manufacturing of small cylinders and the analysis is limited to the powder contained within the cylinders. It is conceivable that the powder bed packing is affected by the raking over printed walls in the vicinity of the powder bed and the lasering process. For example, interactions between the recoater blade and the printed walls such as friction can push powder further in the recoater direction than without the presence of the previously printed layers. The magnitude of these effects depends on the recoater-printed layer interactions, the recoater, and the blade design. Other methods that analyze powder beds without printed sections might be warranted such as in-situ tomography approaches.

The experimental measurement of packing density and porosity is a reference point for the results of the DEM simulations. The effects of different locations along the spreading direction over the build plate on particle packing density in different powder layers are shown in Figure 21a. The powder was spread over the build plate by moving the spreader along the positive Y axis, as shown in Figure 1. The porosity of the powder bed was measured at ten locations along the spreading direction. Two trends can be observed in Figure 21a: as the number of layers increases, the powder bed porosity decreases and eventually approaches a limit, which is about 44% to 45%. The powder bed porosity decreases sharply at the start of the spreading and for the first two to three layers continues to decrease overall along the entire spreading direction. The porosity of layers four and higher remains approximately constant after the initial drop. This change in porosity behavior can be regarded as an effect of the build plate. Only the first layer has direct contact with the build plate, layers two and three are spread over a powder bed but are sufficiently close to the build plate, and the first layer that the effect of the build plate still affects their characteristics. With an increasing number of layers, the powder spreads over previous powder layers and the powder bed characteristics should then become constant if the spreading process itself does not change. The average particle porosity of all deposited layers at different locations along the spreading direction is shown in Figure 21b.

#### 3.2.3. Effects of Spreader Velocity on Powder Particle Packing Density

The effects of increasing the spreader velocity on the powder particle packing density for a powder bed with a 60 µm layer thickness are shown in Figure 22. When the spreader velocity increases, the powder packing density decreases of the single layer that was spread out. At the highest velocities tested in the simulations, between 300 mm/s and 500 mm/s, the packing density and porosity remain approximately constant. With increasing spreader velocity, the interaction time between the spreader and powder decreases, leaving the powder less time to rearrange. The increasing spreader velocity furthermore accelerates the particles to reduce the number of particles that remain on the build plate with some of them being pushed right off the build plate. The recoater velocity was found to be statistically significant for the percent coverage of powder on a build plate and the rate of change of the avalanching angle [31]. Le and coworkers measured the surface particle density of a powder bed as a function of blade material and recoater velocity [32]. The surface density increases with metal recoater velocity in [32] from 0.2 mm/s to 8 mm/s, then remains approximately constant for velocities between 10 mm/s and 80 mm/s before decreasing at higher velocities. The recoater velocities in the current study overlap with Le’s work at recoater velocities of 100 mm/s and 150 mm/s, the highest recoater velocity examined in [32]. In this velocity range the current work agrees with Le’s work in that the particle density decreases with increasing recoater velocity and hence the porosity in the powder bed increases as shown in Figure 22. DEM simulations were performed for powder spreading and the study examined amongst others the effect of the recoater velocity [33]. The same qualitative behavior was observed in this work for a velocity between 25 mm/s and 125 mm/s, the porosity increased. However, the porosity increase amounted only to 5% for a change in velocity of a factor of five. The current work reflects a significantly stronger dependence of the porosity on the recoater velocity. Detailed studies are necessary to shed unambiguous light on the recoater velocity dependence on porosity. Several additional factors are expected to play a role: the powders’ packing behavior, possibly quantifiable by the difference between apparent and tap densities, the atmosphere and humidity but also the recoater tip shape (round, superelliptical, square, etc.). The powder drop at the tip of the powder pile ahead of the recoater is largely gravity-driven. The gravity force remains constant and does not scale with the recoater velocity. It could be argued that with increasing velocity the powder simply becomes more spread out over the build plate, thus lowering the density and increasing the porosity. This simple argument does not explain why the simulations yield an approximately constant porosity at recoater velocities above 0.3 m/s, indicating that more than one mechanism must be operational for powder spreading.

## 4. Summary and Conclusions

This study determined powder particle size distribution and particle packing density of the powder bed in the AM process based on experiments and DEM simulations. The following is a summary of the work presented:The results of the simulation modeling and experimental methodologies for powder particle size distribution were nearly equal. Powder particle sizes were evenly dispersed in both procedures. At the same particle size range, the maximum volume fraction of the particles was likewise found to be the same. Powder particle size distribution was identical across all layers in the simulation.Along the longitudinal direction of the construct plate, the porosity of the particles in the spread layers decreases. The powder that is deposited at the start of the build plate is largely made up of smaller particles that are not evenly distributed. In addition, the powder particle packing density in the top deposited layers is higher than in the lower layers. Furthermore, particle density is more uniform across the entire build plate in the top layers than in the lower layers.At the end of the build plate, the maximum average density of all dispersed powder layers was found to be around 47%. In the top layer, the highest average packing density for the entire spreading direction was determined to be roughly 55%, which is consistent with the experimental finding.With increasing spreader velocity, the powder packing density of the distributed powder drops. The spreader does not have enough time to evenly distribute powder on the build plate at higher speeds. In addition, the fine powder particles were disseminated initially by the spreader at a lesser velocity. Lower spreader velocity enhances the particle packing density of the powder bed in AM due to these considerations. Reduced table displacement raises packing density up to a certain point. When this range exceeds the size range of the maximum particle volume, the packing density drops along with the table displacement. Furthermore, as the spreader deposits more large particles in the powder bed, an increase in table displacement increases the void fraction of the powder bed.

## Figures and Tables

**Figure 1 materials-16-02824-f001:**
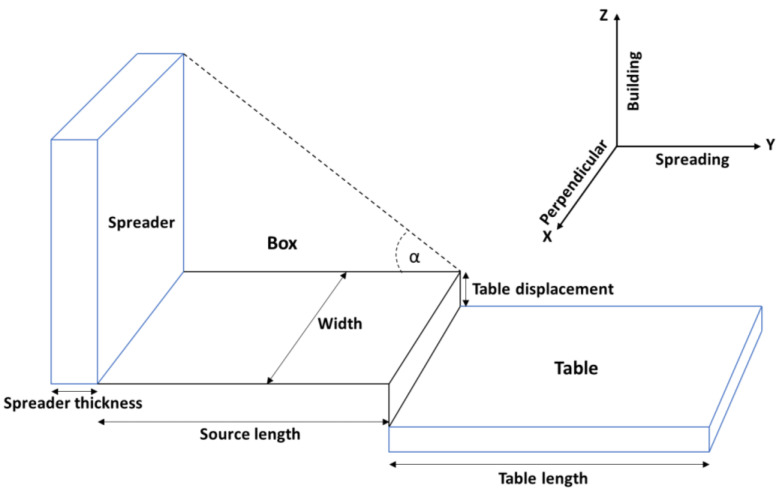
The geometry of the powder spreading system.

**Figure 2 materials-16-02824-f002:**
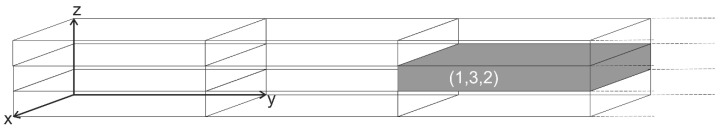
Wireframe depiction of powder bed breakdown into cells. (1,3,2) is indicating the position of the gray cell in (X,Y,Z) coordinates.

**Figure 3 materials-16-02824-f003:**
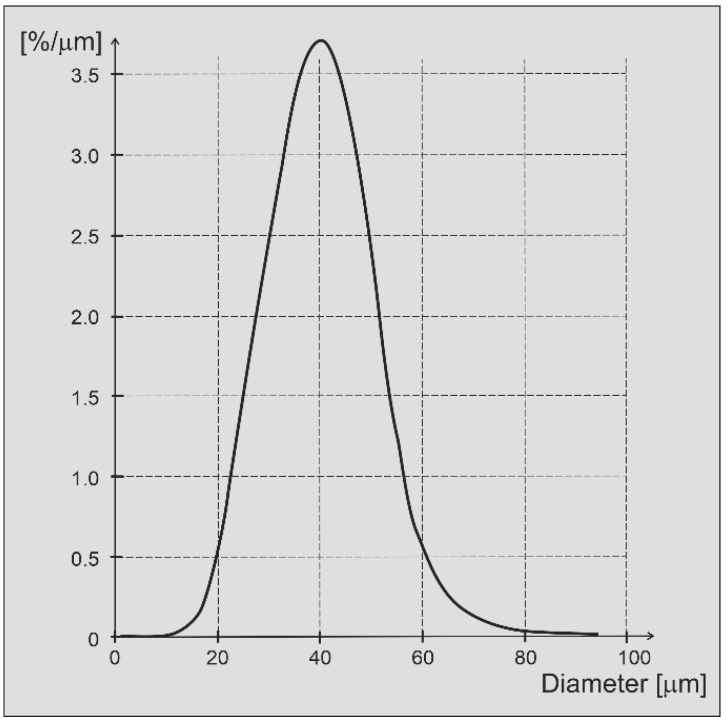
Size distribution of IN718 powder used for experiments.

**Figure 4 materials-16-02824-f004:**
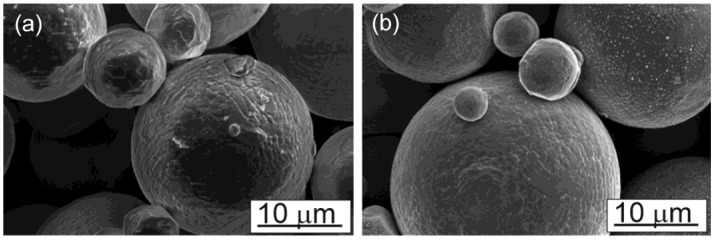
Scanning electron microscopy images of IN718 powder particles were used for this study. (**a**) Particles are spherical in shape. Larger particle appears to be with few satellites. (**b**) Closer view of the particles with smooth spherical surface.

**Figure 5 materials-16-02824-f005:**
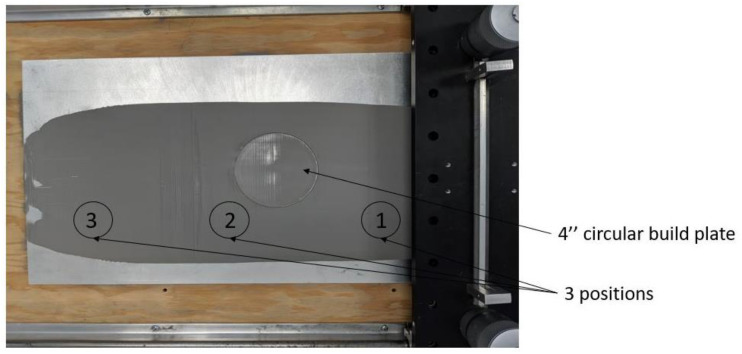
Top view of experimental setup for spreading IN718 powder over a flat plate. Spreading direction right to left. The powder was removed from locations 1–3, which represent an overall distance of 300 mm.

**Figure 6 materials-16-02824-f006:**
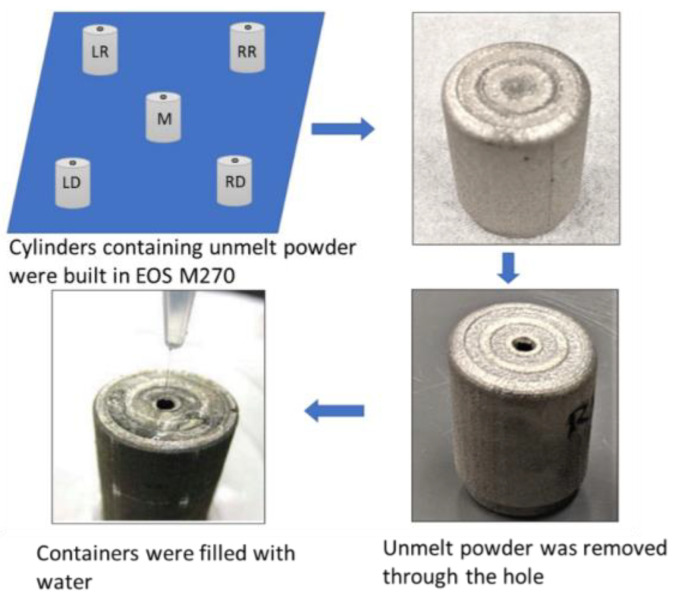
Overview of experimental powder bed packing density measurement.

**Figure 7 materials-16-02824-f007:**
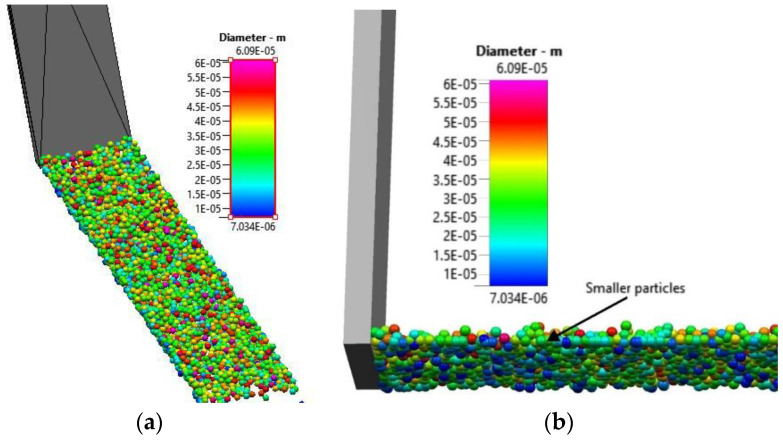
Close view of particles on source table. (**a**) Upper layer of the powder heap (**b**) View of the underside of the powder heap.

**Figure 8 materials-16-02824-f008:**
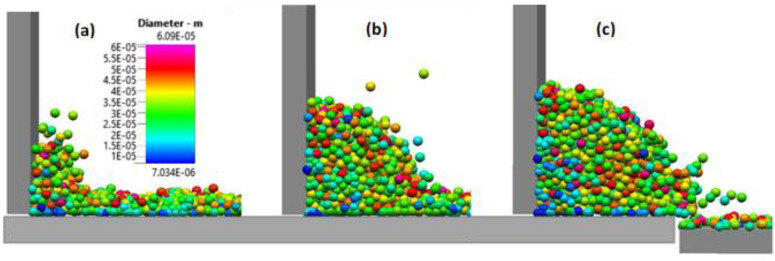
(**a**) Powder pile in front of the spreader at the start of the spreading over the source table. (**b**) Powder pile in front of the spreader halfway along the source table. (**c**) Powder pile in front of the spreader at the end of the source table, transitioning to the build plate.

**Figure 9 materials-16-02824-f009:**
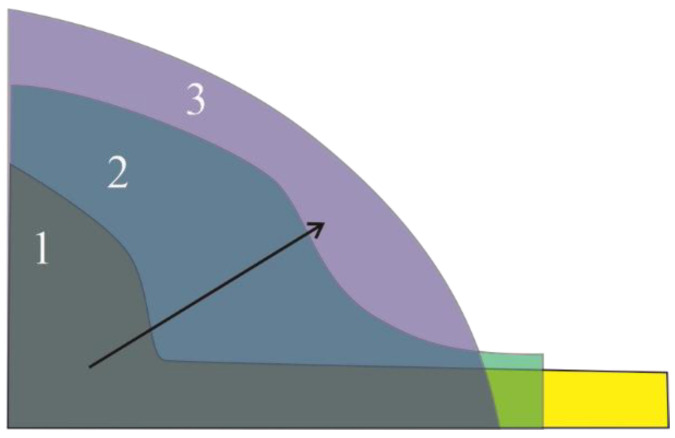
The sequence of powder pile envelop during the spreading over source table. Numbers 1, 2, 3 are indicating the three different positions of the powder pile on the source table as shown in Figure 8a–c respectively.

**Figure 10 materials-16-02824-f010:**
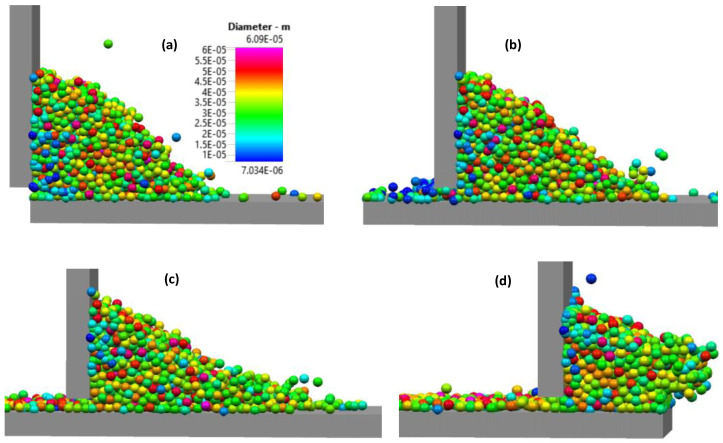
Particle spreading on the build plate. (**a**) At the moment when the rake starts to move over the build plate. (**b**) During the initial raking phase. (**c**) Approximately at the middle of the 4 mm spreading distance. (**d**) At the end of the spreading process over the build plate; particles are pushed on top of the build chamber bottom at the right, which is elevated relative to the build plate by the table displacement.

**Figure 11 materials-16-02824-f011:**
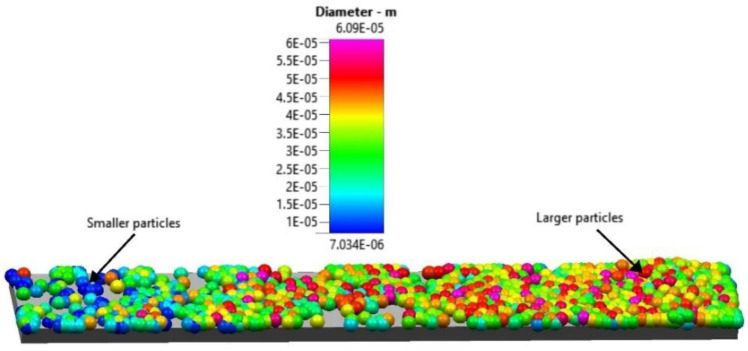
One layer of powder spread over the build plate.

**Figure 12 materials-16-02824-f012:**
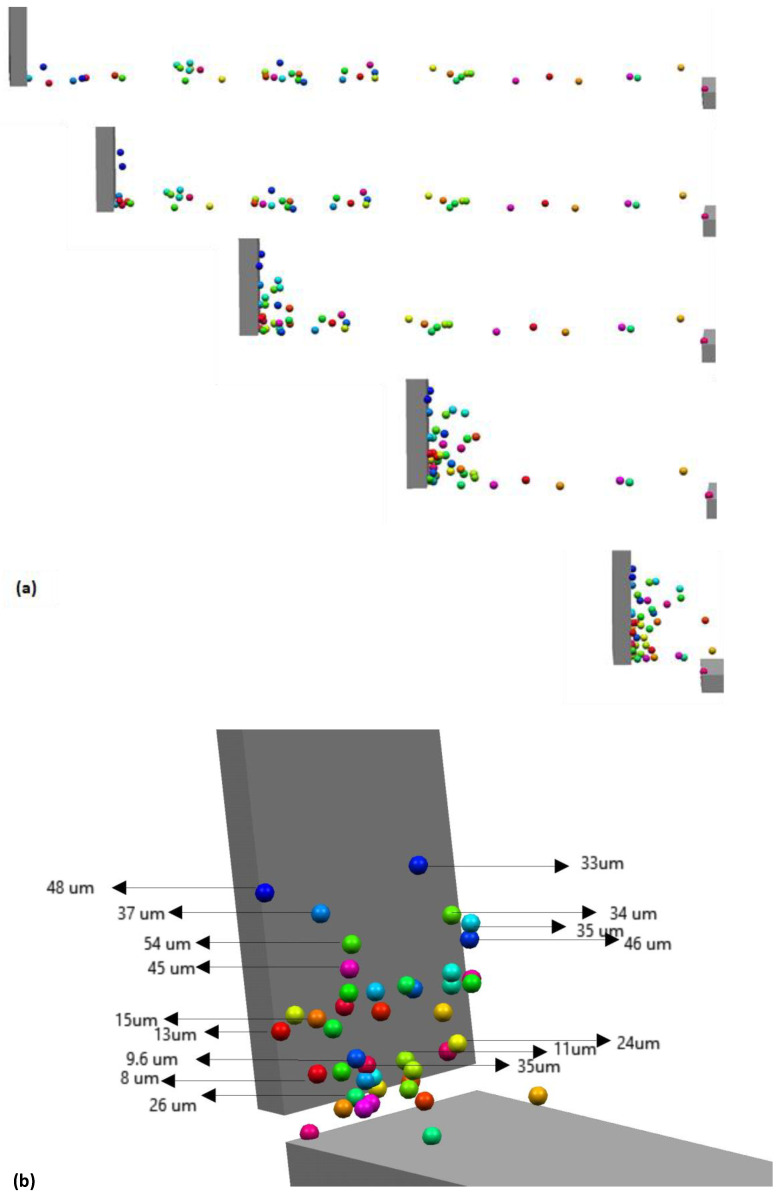
(**a**) The sequence of powder raking over the source plate with 43 particles is highlighted. (**b**) Particle location and sizes at the end of the spreading over the source table.

**Figure 13 materials-16-02824-f013:**
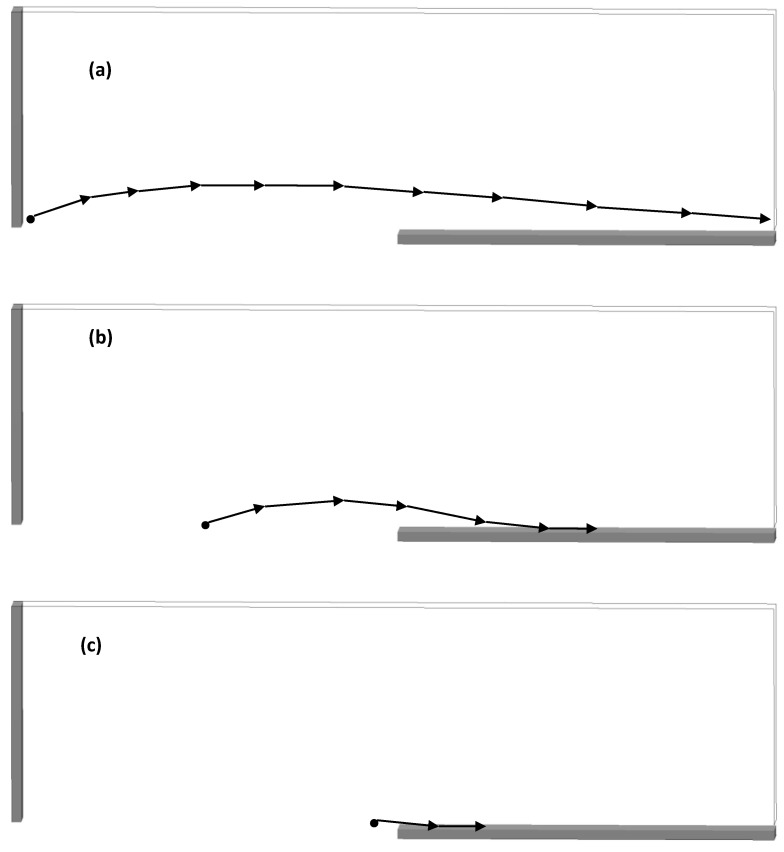
(**a**) The trajectory of individual 49 µm diameter particle across the source and spreader table that was located near the blade at the beginning of the blade motion. (**b**) Same as (**a**) but the particle was located halfway along the source table initially. (**c**) Same as (**a**) but the particle was located toward the right end of the source table initially.

**Figure 14 materials-16-02824-f014:**
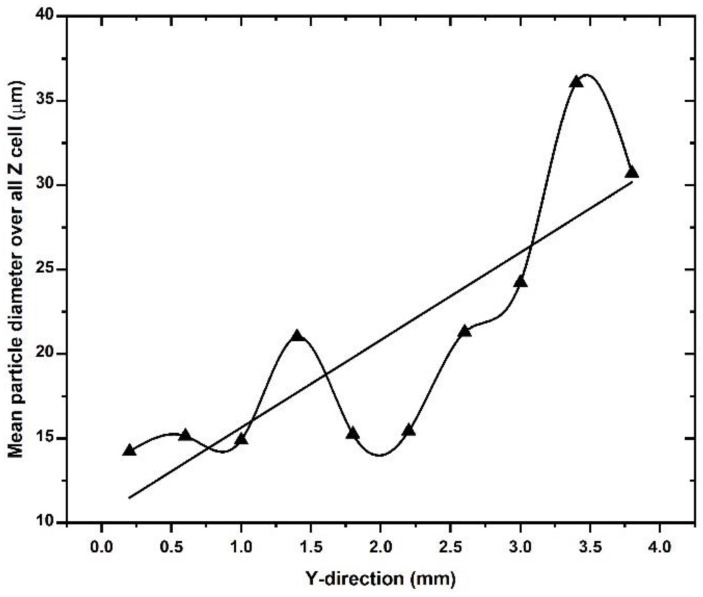
Mean particle size average over all z-cells as a function of the spreading direction (y-direction).

**Figure 15 materials-16-02824-f015:**
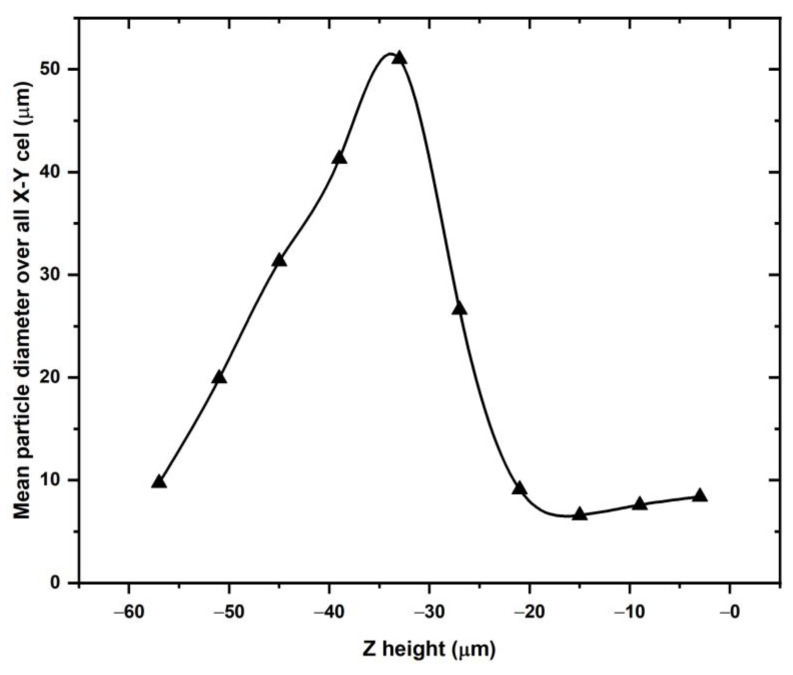
Mean particle size average overall x-y-cells as a function of the z-direction.

**Figure 16 materials-16-02824-f016:**
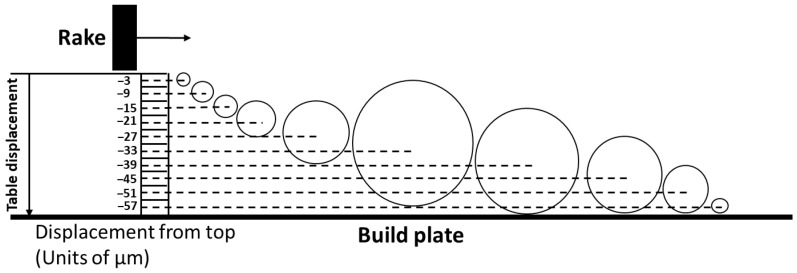
The mean particle size diameter for each cell in the z-direction plotted to scale according to the data from Figure 15.

**Figure 17 materials-16-02824-f017:**
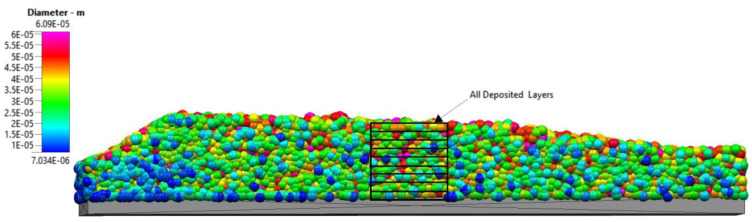
Nine layers of powder deposited over the build plate.

**Figure 18 materials-16-02824-f018:**
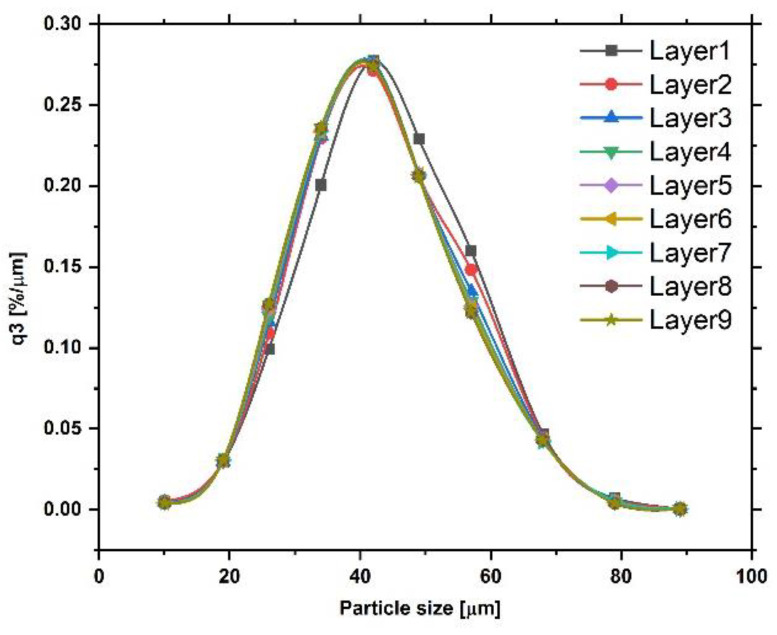
Powder particle size distribution of IN718 powder on the build plate for each of the nine layers that were simulated.

**Figure 19 materials-16-02824-f019:**
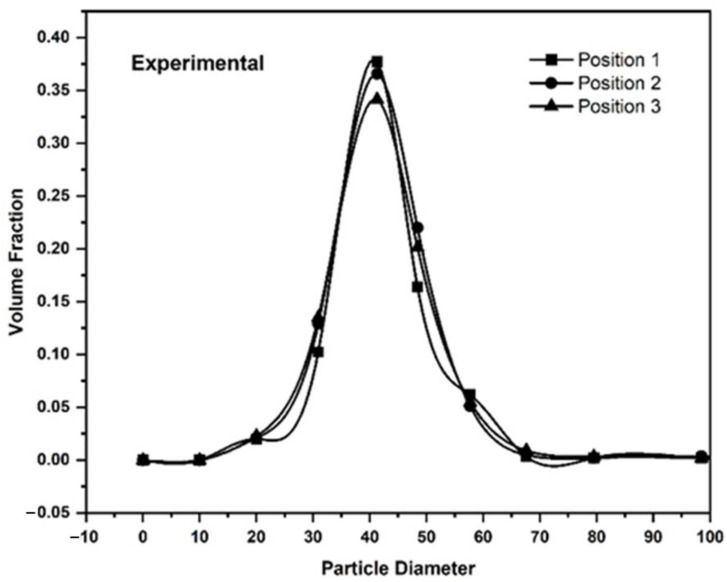
Size distributions of powder taken from the experimental raking device at positions 1–3.

**Figure 20 materials-16-02824-f020:**
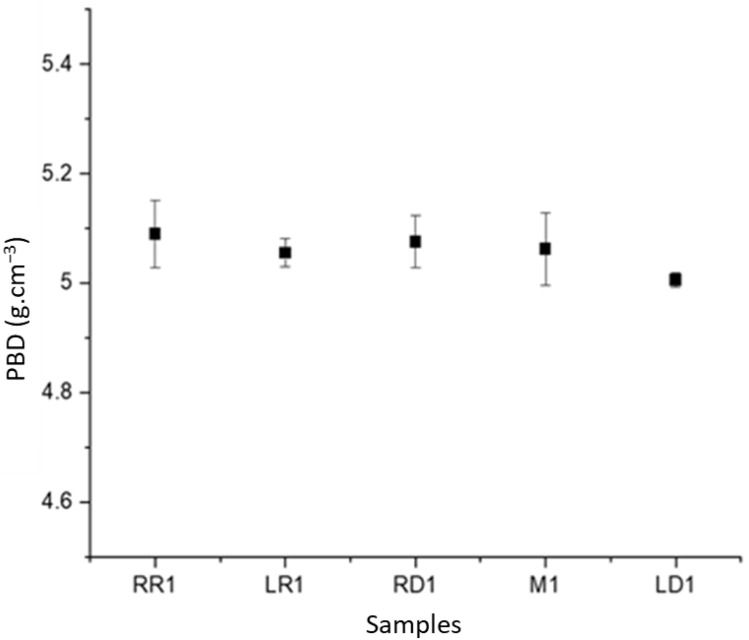
Powder bed density at different locations on the build plate and error bars. RR = right rear, LR = left rear, RD = right front, M1 = center, LD = left front.

**Figure 21 materials-16-02824-f021:**
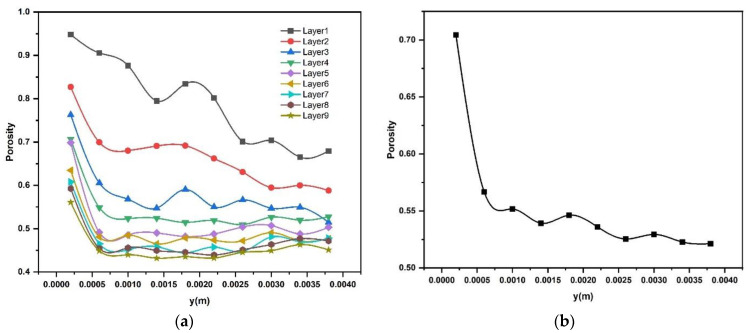
Effect of longitudinal positions of the build plate on the powder particle porosity (**a**) in different deposited layers (**b**) average porosity of powder bed in all layers.

**Figure 22 materials-16-02824-f022:**
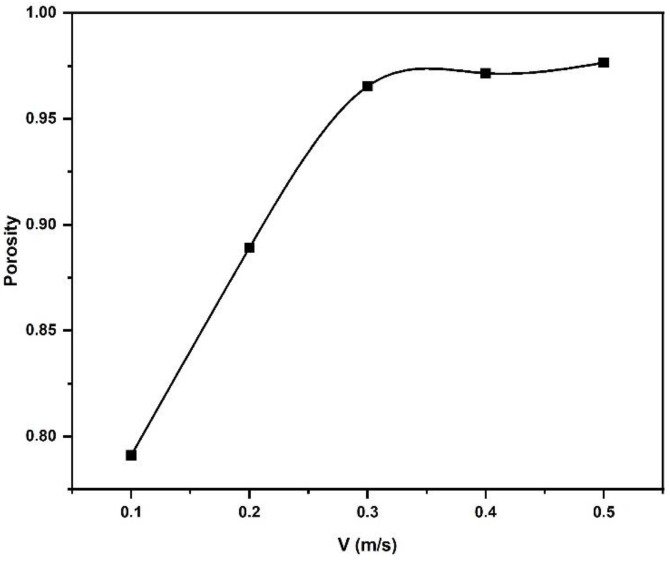
Effect of spreader velocity on the powder bed porosity.

**Table 1 materials-16-02824-t001:** Input parameters for DEM calculations.

Input Parameter	Recommended Value
Material name	IN718
Solid density (kg/m^3^)	8000
Spreader thickness (m)	10^−4^
Source length (m)	4 × 10^−3^
Table length (m)	4 × 10^−3^
Table width (m)	5 × 10^−4^
Table displacement (m)	6 × 10^−5^
Wall thickness (m)	1 × 10^−4^
K-spring coefficient	0.1
Poisson coefficient	0.3
Sliding coefficient	0.1
Rolling coefficient	0.7
Twisting coefficient	0.7
Rolling damping coefficient	1.5
Twisting damping coefficient	1.5
Dosing	0.5
Spreader starting time (s)	2 × 10^−3^
Spreader velocity (m/s)	0.1

## Data Availability

Data is unavailable due to privacy or ethical restrictions.

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
