# Peer review of "Powder Spreading Mechanism in Laser Powder Bed Fusion Additive Manufacturing: Experiments and Computational Approach Using Discrete Element Method"

_materials, 2023, doi:10.3390/ma16072824_

Round 1

Reviewer 1 Report

The manuscript deals with powder spreading mechanism in AM using LBF method. The problem of optimizing the feeder is related to the fluidity of the powder, which depends on the geometric shape and size distribution of the particles. The working speed of the feeder must take into account the technological properties of the powder (apparent density, tap density, flowability) being processed. These aspects of the process should be discussed in the manuscript.

- The method of measuring the particle size distribution is not defined.

- The experimental material is IN718 is not characterized. In the context of the manuscript, I suggest to provide details about geometrical characteristics and surface roughness of the powder particles. SEM documentation can be the best solution.

- I suggest to provide all relevant properties (physical and technological) of the experimental powder.

- The term "powder particle density" gives the impression that it is the internal density of a powder particle. I recommend correcting the terminology.

Reviewer 2 Report

The paper requires a few improvements, but the overall approach is commendable. I am happy to have read it and appreciate the effort put into it.

Good ideas, DEM can be a good method for highly anisotropy Materials, like those of LPBF. I remind a recent work about that On the relationship between cutting forces and anisotropy features in the milling of LPBF Inconel 718 for near net shape parts, International Journal of Machine Tools and Manufacture 170, 103801 and DEM can be the logical system for analyzing. Please discuss the topic anisotropy, include the work and make some comment.  Ref 9 is too general.

The discrete element method (DEM) is a numerical technique used in computational mechanics to simulate the behavior of a large number of discrete particles, such as granular materials or powders, as they interact with each other and with their surroundings.

In the DEM, the particles are represented as discrete bodies, and their interactions are modeled using contact laws and other physical laws. The method is particularly useful for studying complex systems that exhibit non-linear behavior, such as the flow of grains in silos or the behavior of rock masses in mining or geological applications.

Regarding not only one piece but several are usually printed on each plate. The authors also studied the set in Stiffening near-net-shape functional parts of Inconel 718 LPBF considering material anisotropy and subsequent machining issues

Defects are  very little but very dangerous, as it was defined by Coro in A methodology to evaluate the reliability impact of the replacement of welded components by additive manufacturing spare parts, Metals 9 (9), 932 taking this into account, can you include some defects using the DEM. A particle without mass for instance. What do you think?

Why some references are in CAPITAL LETTERS?

The behavior of granular materials can be highly sensitive to the input parameters used in the DEM simulations, such as particle size, shape, and surface properties. Small changes in these parameters can lead to significant changes in the macroscopic behavior of the system, which can make it difficult to obtain accurate and reliable results. Did you do a sensitivity plan for the testing? I think so: PLEASE MAKE BETTER THE ORGANIZATION OF SECTION, SUBSECTIONS…

A manual powder spreading system was employed: ok. Please define if you really printed the components. I think the density and distribution could be changed by thw whipper and the laser. Please comment. The cylinder were placed in good spots, but the whipper(sweeper) can push poweder to the sides.

 Would it be possible for you to kindly dedicate some extra effort in refining the paper and submitting it at your earliest convenience? I would be most grateful for your hard work and dedication.
